# Nutrition Therapy Promotes Overall Survival in Cachectic Cancer Patients through a New Proposed Chemical-Physical Pathway: The TiCaCONCO Trial (A Randomized Controlled Single-Blinded Trial)

Elisabeth De Waele [1,2,*,†], Joy Demol [1,2,†], Koen Huysentruyt [2,3], Geir Bjørklund [4], Ronald Buyl [5], Alessandro Laviano [6] and Joeri J. Pen [2,7,*]

1 Intensive Care Unit, UZ Brussel, Vrije Universiteit Brussel (VUB), 1090 Brussels, Belgium
2 Department of Nutrition, UZ Brussel, Vrije Universiteit Brussel (VUB), 1090 Brussels, Belgium
3 Department of Pediatric Gastroenterology, UZ Brussel, Vrije Universiteit Brussel (VUB), 1090 Brussels, Belgium
4 Council for Nutritional and Environmental Medicine, 8610 Mo i Rana, Norway
5 Department of Public Health, Biostatistics and Medical Informatics Research Group, Vrije Universiteit Brussel (VUB), 1090 Brussels, Belgium
6 Department of Translational and Precision Medicine, University of Rome La Sapienza, 00185 Rome, Italy
7 Diabetes Clinic, Department of Internal Medicine, UZ Brussel, Vrije Universiteit Brussel (VUB), 1090 Brussels, Belgium
* Correspondence: elisabeth.dewaele@uzbrussel.be (E.D.W.); joeripen@gmail.com (J.J.P.)
† These authors contributed equally to this work.

**Abstract:** Cancer threatens nutritional status, and many patients will become cachectic with a negative impact on prognosis. In the TiCaCo pilot trial, we showed a positive effect of calorie matching Nutrition Therapy on both morbidity and mortality. We attempt to validate these results in the TiCaCONCO trial. In a prospective, randomized, single-blinded, controlled trial, patients were treated with either intensive, individual biometric parameter-oriented dietary counseling (nutrition therapy or NT) for a maximum period of three months, or regular dietary counseling (control or CT), before and during conventional cancer treatment. Sixty patients were enrolled over a two-year period, with 30 receiving nutrition therapy and 30 being controls. The primary endpoint was overall survival (OS). Overall survival at 12 months in all patients was 47% (14/30 patients) in the CT group with a median OS of 45.5 weeks, and 73% (22/30 patients) in the NT group with a median OS that was undefined (i.e., cannot be calculated, as >50% of patients in the NT group were still alive at the end of the study) ($p = 0.0378$). The survival difference still exists when only male patients are analyzed, but is not observed in female patients. Biophysical measurements were performed at 0, 3, and 12 months in all patients. In men, the differences between CT vs NT were statistically significant for body hydration ($p = 0.0400$), fat mass ($p = 0.0480$), total energy expenditure ($p = 0.0320$), and median overall survival at 12 months ($p = 0.0390$). At 3 months (end of the intervention), the differences between CT vs NT for body hydration were 73 ± 3% vs. 75 ± 5%, for fat mass 14 ± 4% vs. 19 ± 5%, and for total energy expenditure 2231 ± 637 Kcal vs. 2408 ± 369 Kcal. In women, the differences between CT vs NT were not statistically significant for body hydration ($p = 1.898$), fat mass ($p = 0.9495$), total energy expenditure ($p = 0.2875$) and median overall survival at 12 months ($p = 0.6486$). At 3 months (end of the intervention), the differences between CT vs. NT for body hydration were 74 ± 2% vs. 78 ± 5%, for fat mass 25 ± 7% vs. 29 ± 19%, and for TEE 1657 ± 297 Kcal vs. 1917 ± 120 Kcal. Nutrition Therapy, based on patient-specific biophysical parameters, including the measurement of metabolism by indirect calorimetry and body composition measurements by BIA, improves overall survival, at least in men. The mechanism would be increasing extra energy for the body, which is necessary to fight off cancer.

**Keywords:** cachexia; cancer; mortality; nutrition; survival; biophysics

## 1. Introduction

Cancer is a globally spread disease with high prevalence and incidence, often of poor outcome because it is typically diagnosed at a late stage. One reason for this prognosis is the presence of malnutrition, both because of an impaired intake (due to anorexia) and because of malignancy-induced hyper-catabolism. This causes the appearance of a special form of cachexia, named cancer-associated cachexia, and is (usually) defined as a weight loss more than or equal to five percent during the six months prior to the time of diagnosis. Nutritional intervention in these patients had already been shown to decrease morbidity, to increase progression-free survival, and to increase wellbeing. Our pilot study was the first to show that nutritional intervention—making use of the ESPEN (European Society for Clinical Nutrition and Metabolism) directives, bioelectrical impedance analysis (BIA), and indirect calorimetry—might increase overall survival. We therefore wanted to validate these preliminary results in a more elaborated RCT (randomized controlled single-blind trial) to investigate whether nutritional intervention could be considered as a new cancer treatment modality in its own right, instead of being merely a supportive action [1–6].

In our pilot study, twenty patients were randomized; ten received regular counseling by dietitians, while the other ten received intensified nutrition therapy based on practical measurements of caloric needs, rather than theoretical calculations. In the interventional group, a measurement of biophysical parameters was done (including the Bioelectrical Impedance Analysis or BIA for body composition), and the patients' energy expenditure was measured with indirect calorimetry. Based on these findings, the patients received nutritional interventions according to the ESPEN guidelines. Supplementary interventions were made to match the caloric intake to the actual energy expenditure, making use of enteral and/or parenteral nutrition when indicated. This was done with the method of an intensive coaching and follow-up to continue this nutrition strategy (with dieticians being "on call" after normal working hours). Despite the study follow-up lasting for two years, nutritional intervention only took place during the first three months. The results were rather baffling: the patients of the intervention group maintained their body weight, they experienced far fewer unplanned hospitalization days, and they also seemingly lived much longer [7–13].

## 2. Materials and Methods

We conducted a prospective, randomized, controlled, single-blinded trial in the University Hospital Brussel (UZ Brussel), Belgium: The Tight Caloric Control in Oncologic Patients (TiCaCONCO) trial. The study was approved by the Institutional Review Board of the hospital and performed in accordance with the Declaration of Helsinki and Good Clinical Practice guidelines. Written informed consent was obtained from all patients. Protocol number: NCT03058107.

This trial targeted the validation of the TiCaCo [10] results. Patients with cancer types where cachexia is frequently seen were included. Weight loss could also manifest itself later on, and was not always present at the start of the study (contrary to Fearon's more stringent criteria [12]). The patients were randomized using a closed-envelope system. The recruitment phase lasted 2 years, while the follow-up phase lasted 1 year (after actual inclusion), but nutritional intervention only took place during active oncological treatment (3 months for chemotherapy, 6 weeks for radiotherapy). The protocol was amended three times to allow for a more successful enrolment. A step-by-step explanation and flowchart can be found in the Protocol (Supplementary Material).

The primary outcome was overall survival at 12 months. The secondary outcomes were hospitalization, morbidity, weight stabilization, body composition, energy expenditure, and complete remission. However, due to missing data, not all secondary endpoints could be assessed properly.

Control Therapy is standard nutritional counseling by dietitians specialized in oncology. Energy expenditure was only measured (and not used) in this standard protocol. The patients were screened at diagnosis and received standard dietary counseling for 1 year

when assigned to group B. Dietary intervention implied oral, parenteral, and/or enteral nutrition, depending on the patient's general status.

Nutrition Therapy is intensive dietary counseling based on practical measurements of energy expenditure, in contrast to the calculation of theoretic formulas (most often Harris–Benedict), or lack of any scientific method altogether. The patients were screened at diagnosis and received Nutrition Therapy when assigned to group A. The actual interventional period remained for 6 weeks to 3 months (the duration of the oncological intervention), after which normal dietary counseling was provided. Dietary intervention implied oral, parenteral, and/or enteral nutrition, depending on the patient's general status, but with the specific goal of matching caloric intake with resting energy expenditure.

Briefly, the goal was to restore the daily and cumulated energy balance, limiting the caloric deficit (intake to need deviation) to a maximum of 50%. The caloric value of intake was calculated by the dietician using the national dietary software program Nubel (asbl). Whenever the total amount of calories could not match 60% of the caloric need for whatever reason, a nutritional intervention, according to the ESPEN guidelines, was launched by the study dietitian. The weekly assessment included caloric need, caloric intake, energy deficit, presence of a nutritional intervention, type of intervention and artificial feeding, and proportion of each feeding type within the total amount of caloric intake.

### 3. Statistical Analysis

The statistical power was calculated using an online tool (ClinCalc) based on overall survival at 12 months in our previous TiCaCo trial (due to the lack of any literature), showing a need for only 60 patients (48 + 12 patients, anticipating 20% dropouts, with 95% power). The statistical analysis was performed using GraphPad Prism 9.

The Kaplan–Meier approach was used for estimating the overall survival. Patients who were withdrawn or became lost to follow-up were censored at the date of last visit or at the last date of study medication, whichever occurred later. A Mantel–Cox test was used for survival analysis.

Continuous data were assessed with an unpaired, two-tailed *t*-test, using treatment as an independent variable, to compare the baseline parameters. Numerical variables were assessed with the Fisher Exact test. Repeated measures were assessed with a mixed effects model, also using Greenhouse–Geisser correction, as some data were missing (e.g., mortality, dropout, or patients not wearing the SenseWear at home).

Finally, simple linear regression analysis was performed (biophysical parameters with overall survival as variable parameter). Analyses were only performed in men, due to the previous results. The analysis should be interpreted with caution, as some data are missing.

### 4. Results

The patients' summarized neoplasia characteristics are provided in Table 1.

**Table 1.** Neoplastic characteristics per patient group.

|  | CT | NT |
|---|---|---|
| Non-Small Cell Lung | 15 | 12 |
| Head and Neck | 4 | 8 |
| Oesophageal | 4 | 3 |
| Pancreatic | 1 | 1 |
| Colorectal | 3 | 3 |
| Small Cell Lung | 2 | 1 |
| Unknown Primary | 1 | 2 |

The patients' general characteristics at baseline can be found in Table 2.

**Table 2.** Patient characteristics at baseline.

| | CT | NT |
|---|---|---|
| Age (years) | 65.5 ± 10.1 | 58.9 ± 10.6 |
| BMI (kg/m$^2$) | 23.69 ± 3.65 | 24.64 ± 4.34 |
| M/F ratio | 20/10 | 18/12 |
| Hb (g/dL) | 13.2 ± 5.9 | 12.7 ± 5.6 |
| WBC (cell count × 10$^3$/mm$^3$) | 8.6 ± 3.4 | 8.0 ± 3.4 |
| Creatinin (mg/dL) | 0.76 ± 0.26 | 0.84 ± 0.26 |
| Albumin (g/L) | 38 ± 5 | 38 ± 5 |

The inclusion and exclusion criteria are shown in Table 3.

**Table 3.** Inclusion and exclusion criteria.

| Inclusion | Exclusion |
|---|---|
| >18 years | Concomitant second malignancy |
| Male and female | Uncertainty of diagnosis |
| Colorectal, lung, esophageal, gastric, pancreatic or head and neck cancer before chemo- or radiotherapy is started (naive to treatment), but surgery may already have been performed OR relapse >3 months after initial oncologic therapy | Patient unfit for chemotherapy, radiotherapy, or surgery |
| Oncologic cachexia (undesired weight loss >5% in less than 6 months), before or during treatment | Palliative treatment or terminal patient (life expectancy <3 months) |
| Written informed consent/ability to give informed consent | Patient already participating in another study |
| | Pregnancy/lactation |
| | Any other pathology present that causes the patient to be unfit for oncologic therapy (e.g., end-stage renal failure, severe chronic obstructive pulmonary disease, severe heart failure) |
| | Unable to adhere to protocol instructions (e.g., language barrier) |
| | Investigator's uncertainty about the willingness or ability of the patient to comply with the protocol requirements |
| | Participation in any other studies involving investigational or marketed products concomitantly or within two weeks prior to entry into the study |

The median overall survival (OS) in the CT population was 45.5 weeks. The Kaplan–Meier OS estimates were 14/30 (47%) patients at 12 months. The median OS in the NT population was undefined (i.e., cannot be calculated, as >50% of patients in the NT group were still alive at the end of the study). The Kaplan–Meier OS estimates were 22/30 (73%) patients at 12 months (Figure 1). The difference in overall survival was statistically significant ($p$ = 0.0378). In all deceased patients, mortality was directly or indirectly related to oncological status.

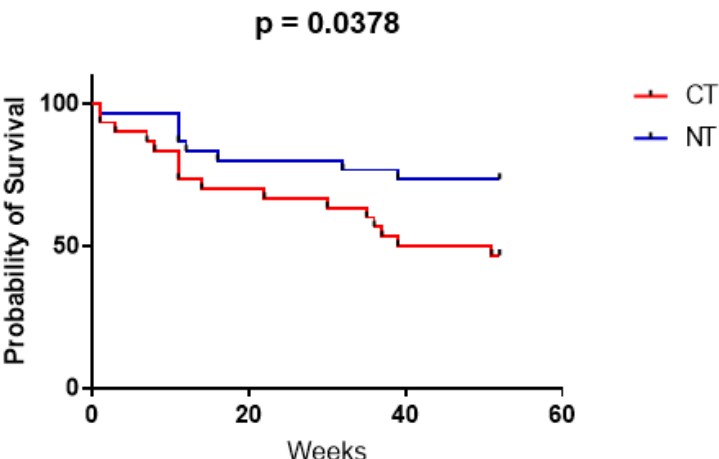

**Figure 1.** Kaplan–Meier curves showing patient survival over 12 months, for both the CT (red) and NT (blue) group, calculated with the Mantel–Cox test (GraphPad Prism 9).

Figure 2A1,B1,C1,D1: In men, the differences between CT vs. NT were statistically significant for body hydration ($p$ = 0.0400), fat mass ($p$ = 0.0480), total energy expenditure ($p$ = 0.0320), and median overall survival at 12 months ($p$ = 0.0390). At 3 months (end of the intervention), the differences between CT vs. NT for body hydration were 73 ± 3% vs. 75 ± 5%, for fat mass 14 ± 4% vs. 19 ± 5%, and for TEE 2231 ± 637 Kcal vs. 2408 ± 369 Kcal. Figure 2A2,B2,C2,D2: In women, the differences between CT vs. NT were not statistically significant for body hydration ($p$ = 1.898), fat mass ($p$ = 0.9495), total energy expenditure ($p$ = 0.2875), and median overall survival at 12 months ($p$ = 0.6486). At 3 months (end of the intervention), the differences between CT vs. NT for body hydration were 74 ± 2% vs. 78 ± 5%, for fat mass 25 ± 7% vs. 29 ± 19%, and for TEE 1657 ± 297 Kcal vs. 1917 ± 120 Kcal. Biophysical measurements were performed at 0, 3, and 12 months in all patients.

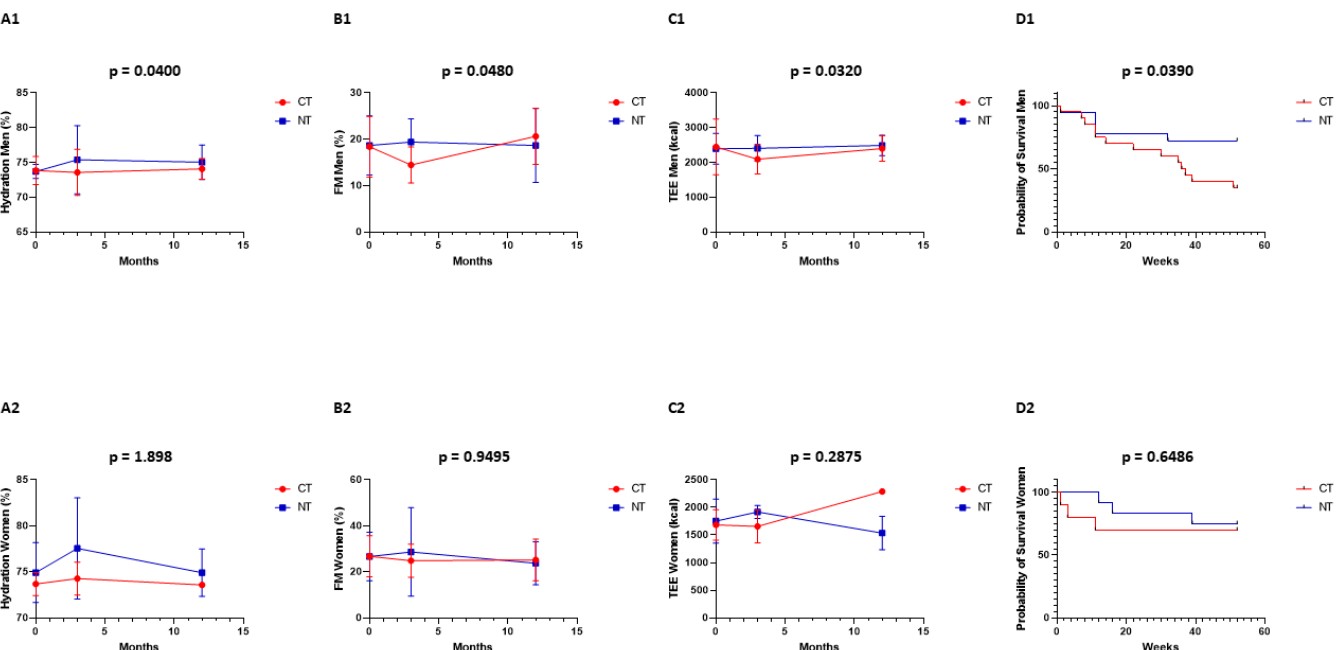

**Figure 2.** Biophysical measurements were performed at 0, 3, and 12 months in all patients: control (CT, red) and interventional (NT, blue) group. Men: (**A1**) hydration, (**B1**) fat mass, (**C1**) total energy expenditure, and (**D1**) overall survival. Women: (**A2**) hydration, (**B2**) fat mass, (**C2**) total energy expenditure, and (**D2**) overall survival.

Figure 3 shows simple linear regression analysis in men for body hydration (A), fat mass (B), and total energy expenditure (C). The variable parameter is overall survival.

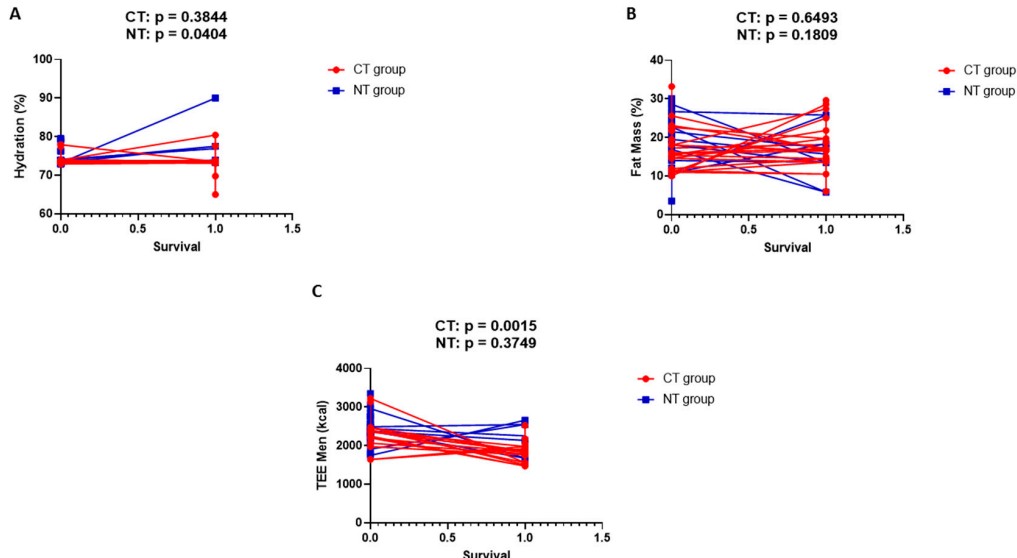

**Figure 3.** Simple linear regression analysis in men for body hydration (**A**), fat mass (**B**), and total energy expenditure (**C**). The variable parameter is overall survival.

## 5. Discussion

This study shows that Nutrition Therapy appears to be capable of improving overall survival (OS) in cancer altogether, although this is only observed in male patients. Our original pilot TiCaCo trial had already suggested an advantage concerning mortality overall, but a mechanism to explain the observed features had not been investigated. As such, the clinical results in this small trial could be considered reliable. Our use of biophysical measurements, rather than biochemical calculations, might be the reason why we succeeded where others had failed before.

When applying Nutrition Therapy, we notice an increase in both hydration and fat mass, both of which are well known to equally increase the energetic capacity of the human body. Along with an elevated TEE, these features would allow the human body to save more energy to fight off cancer, although we could not investigate how exactly this energy would be used. It should be noted that studies on indirect calorimetry in cancer are relatively rare, leading to guidelines based on consensus [14].

Altogether, a new pathway is proposed, with the technique making use of both physical chemistry and chemical physics. Strikingly, these results were only observed in male patients, where the regression analysis confirmed the improved hydration. This could very well indicate that body hydration is a more important therapeutic target than any other nutrient is. In contrast, none of these results can be observed in women, although there are too few female patients to draw definite conclusions. The BIA's role in cancer has been relatively well documented, but the results on different parameters may indeed be contradictory [15–20].

An important drawback would be the small sample size, which does not allow the performance of any far-driven statistics. Moreover, statistical power was only calculated for overall survival between the two groups, and not for men and women apart, nor for any other parameter. The statistical power calculation is also entirely based on our previous pilot study, as there was no other literature to be found. There is also some heterogeneity, since multiple cancers were examined together, being a direct consequence of multiple protocol amendments, and given the small difference in age between the two groups. Simple linear regression analyses were only performed in men, but the analysis should be interpreted with caution, as some data are missing. As such, this trial cannot fully confirm

the results of our previous trial, but the results do remain the same. Therefore, despite the lack of statistical soundness seen in much larger trials, this study does provide high hopes for patients with advanced-stage cancer.

In conclusion, cancer-related cachexia could be effectively counteracted by a patient-tailored, measurement-based nutritional approach, under the supervision of a dedicated multidisciplinary nutrition team, and making use of advanced biophysical parameters (especially body hydration). At least in our trial, this resulted in an improved one-year OS. The presence of a likely pathway adds strength to the clinical results. Major advantages are the fairly low cost, especially when compared to biochemical cancer treatments, accompanied by a relatively short learning curve. Together, the results can easily and readily be implemented in international nutritional guidelines.

**Supplementary Materials:** The following information can be downloaded at: https://www.mdpi.com/article/10.3390/j5040032/s1, Table S1: Protocol; Table S2: CONSORT Flow Diagram; Table S3: CONSORT Checklist.

**Author Contributions:** Conceptualization, E.D.W. and J.J.P.; data curation: J.D.; formal analysis: R.B. and J.J.P.; funding acquisition: E.D.W. and J.J.P.; investigation: E.D.W., J.D. and J.J.P.; methodology: E.D.W., J.D. and J.J.P.; project administration: E.D.W., J.D. and J.J.P.; resources: E.D.W. and J.J.P.; software: E.D.W. and J.J.P.; supervision: E.D.W. and J.J.P.; validation: E.D.W. and J.J.P.; visualization: J.J.P.; writing—original draft: E.D.W., K.H., G.B., A.L. and J.J.P.; writing—review and editing: E.D.W. and J.J.P. All authors have read and agreed to the published version of the manuscript.

**Funding:** This research was funded by Baxter and Nutricia (no grant numbers available) and the APC was funded by J.P. (review vouchers).

**Institutional Review Board Statement:** The study was conducted in accordance with the Declaration of Helsinki, and approved by the Ethics Committee of UZ Brussel (protocol code B.U.N. 143201629790 on 7 December 2016).

**Informed Consent Statement:** Informed consent was obtained from all subjects involved in the study, also to publish this paper.

**Data Availability Statement:** Data are available upon request. Preprint: https://zenodo.org/record/6255816#.Y16Hdr3MKM8 (accessed on 19 August 2022). Significance Statement: Nutrition Therapy (NT) improved the overall survival in cancer patients (statistically powered, based on our pilot trial from 2015). The analysis of the differences between the sexes yielded remarkable results. The survival advantage appeared to be solely due to enhanced survival in men, caused by a new biophysical pathway. Basically, NT enhances both hydration and fat mass, causing more total energy expenditure (TEE) to become available. None of this was observed in women. Hypothetically, this leaves more energy to the body to fight off cancer. An actual pathway is evidenced, albeit biophysical, rather than biochemical.

**Acknowledgments:** The authors declare no conflict of interest. This trial was funded by Baxter Healthcare International and Nutricia (Danone), through grants; the research was performed independently. The authors would like to thank the Anticancer Fund (Antikankerfonds) for moral support.

**Conflicts of Interest:** The authors declare no conflict of interest.

## Abbreviations

CT: control group; NT: nutrition therapy group; OS: overall survival; BIA: bioelectrical impedance analysis; TEE: total energy expenditure.

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
