# Peer review of "Nutrition Therapy Promotes Overall Survival in Cachectic Cancer Patients through a New Proposed Chemical-Physical Pathway: The TiCaCONCO Trial (A Randomized Controlled Single-Blinded Trial)"

_2571-8800, doi:10.3390/j5040032_

Round 1

Reviewer 1 Report (Previous Reviewer 3)

Thank you for submitting a revised manuscript. In response to the suggestion that the "Discussion include additional content explaining how the study results and proposed pathway relate to other research in the literature" authors responded that the Discussion was rewritten and citations were provided as needed. However, it does not appear that any citations were included in the revised Discussion.

Author Response

Rebuttal 2

Dear,

Thank you for allowing us to resubmit after the second review round. Below, you can find our answers on the editors's comments & remarks. Altogether, most remaining comments dealt with adding references and formatting to the journal style, which we have taken care of.

Kind regards,

Joeri Pen

Reviewer #1

Thank you for submitting a revised manuscript. In response to the suggestion that the "Discussion include additional content explaining how the study results and proposed pathway relate to other research in the literature" authors responded that the Discussion was rewritten and citations were provided as needed. However, it does not appear that any citations were included in the revised Discussion.

Indeed; some recent citations were included.

Reviewer 2 Report (New Reviewer)

The manuscript entitled: „Nutrition therapy promotes overall survival in cachectic cancer patients through a new proposed chemical-physical pathway: the TiCaCONCO trial (A randomized controlled single-blinded trial)” presents scientific and practical importance in the medical and nutrition fields.

In this manuscript, De Waele et al. demonstrated, from a prospective, randomized, single-blind, controlled study, that nutritional therapy, based on patient-specific biophysical parameters, including metabolic measurements by indirect calorimetry and body composition measurements by BIA, improves overall survival in men. The article is quite interesting but needs improvement.

I have attached several observations that I hope will be useful in improving the manuscript:

- More details should be provided on determining respondents' energy expenditure by indirect calorimetry.

- The manuscript isn’t prepared according to the journal's requirements: the tables shown in the results must be inserted right after their citation in the text. Figures (CONSORT 2010 Flow Chart &Diagram, figure 1, 2, 3)  are missing from the manuscript.

- Abbreviations must be explained when the first citation is.

- Materials and Methods section requires restructuring (there are no references). The protocol number should be provided. A sequence of methodological stages (steps) and a figure summarising the process should be indicated.

-The authors should revise the Discussion; there aren't references to the literature.

- The study isn’t well documented, including only a number of 13 bibliographic references.

- References should be checked for uniformity in the formatting style.

- Editorial correction needed.

Author Response

Rebuttal 2

Dear Editor,

Thank you for allowing us to resubmit after the second review round. Below, you can find our answers on the editors's comments & remarks. Altogether, most remaining comments dealt with adding references and formatting to the journal style, which we have taken care of.

Kind regards,

Joeri Pen

Reviewer #2

The manuscript entitled: „Nutrition therapy promotes overall survival in cachectic cancer patients through a new proposed chemical-physical pathway: the TiCaCONCO trial (A randomized controlled single-blinded trial)” presents scientific and practical importance in the medical and nutrition fields.

In this manuscript, De Waele et al. demonstrated, from a prospective, randomized, single-blind, controlled study, that nutritional therapy, based on patient-specific biophysical parameters, including metabolic measurements by indirect calorimetry and body composition measurements by BIA, improves overall survival in men. The article is quite interesting but needs improvement.

I have attached several observations that I hope will be useful in improving the manuscript:

- More details should be provided on determining respondents' energy expenditure by indirect calorimetry.

In Materials and Methods, a more in-dept part is added, and marked in yellow.

- The manuscript isn’t prepared according to the journal's requirements: the tables shown in the results must be inserted right after their citation in the text. Figures (CONSORT 2010 Flow Chart &Diagram, figure 1, 2, 3)  are missing from the manuscript.

This has been done for the tables. We kept the Figures in the TIFF files, for a better resolution (copy-paste of GraphPad figures in PowerPoint).

- Abbreviations must be explained when the first citation is.

We have taken care of this.

- Materials and Methods section requires restructuring (there are no references). The protocol number should be provided. A sequence of methodological stages (steps) and a figure summarising the process should be indicated.

All necessary information was added in yellow.

-The authors should revise the Discussion; there aren't references to the literature.

References were added to the Discussion, which in itself had already been toned down and could not be further improved, due to the relative lack of existing literature (especially concerning indirect calorimetry in the oncological setting).

- The study isn’t well documented, including only a number of 13 bibliographic references.

A number of (recent) references was added, especially to the Discussion.

- References should be checked for uniformity in the formatting style.

This has been done.

- Editorial correction needed.

The paper was formatted to the correct journal style.

Round 2

Reviewer 2 Report (New Reviewer)

Figures 1, 2, and 3 are still missing in the manuscript:

Figure 1: Kaplan-Meier curves showing patient survival over 12 months, for both the CT (red) and 289 NT (blue) group, calculated with the Mantel-Cox test (GraphPad Prism 9). 290

Figure 2: Biophysical measurements were performed at 0, 3 and 12 months in all patients: control 291 (CT, red) and interventional (NT, blue) group. Men: A1, Hydration. B1, Fat Mass. C1, Total Energy 292 Expenditure. D1, overall survival. Women: A2, Hydration. B2, Fat Mass. C2, Total Energy Expendi- 293 ture. D2, overall survival. 294

Figure 3: Simple linear regression analysis in men for body hydration (A), fat mass (B) and total 295 energy expenditure (C). The variable parameter is overall survival.

Author Response

Figures 1, 2, and 3 are still missing in the manuscript:

Figure 1: Kaplan-Meier curves showing patient survival over 12 months, for both the CT (red) and 289 NT (blue) group, calculated with the Mantel-Cox test (GraphPad Prism 9). 290

Figure 2: Biophysical measurements were performed at 0, 3 and 12 months in all patients: control 291 (CT, red) and interventional (NT, blue) group. Men: A1, Hydration. B1, Fat Mass. C1, Total Energy 292 Expenditure. D1, overall survival. Women: A2, Hydration. B2, Fat Mass. C2, Total Energy Expendi- 293 ture. D2, overall survival. 294

Figure 3: Simple linear regression analysis in men for body hydration (A), fat mass (B) and total 295 energy expenditure (C). The variable parameter is overall survival.

Response: All 3 main figures were pasted in the Results section, by inserting the TIFF-files -- hoping the figure's quality will be sufficient.

This manuscript is a resubmission of an earlier submission. The following is a list of the peer review reports and author responses from that submission.

Round 1

Reviewer 1 Report

Dear Editors and Authors,
I kindly thank you for the opportunity to review.

Broad comments:

This is an RCT about intensive versus conventional nutritional therapy in cachectic cancer patients. The topic is clearly of interest within our research community and the authors are well-renowned. Unfortunately this manuscript lacks quite a lot of detail regarding patients, interventions, data collection, outcomes, statistics and scientific background. Language and structure revisions are needed, the figures are missing. I therefore cannot recommend acceptance at this moment.

This article needs a language revision – many of the phrases are unclear or imprecise – please see my comments below. The article needs more structure – the authors seem to jump between topics several times (see specific comments below). Abbreviations need to be used or defined (table 1).

The abstract would benefit from being more concise and detailed regarding the nutrition therapy and the results of the biomarkers/ BIA/ calorimetry results. Also, the abstract promises a “ new mechanism only in men” which is not properly described in the entire manuscript and abstract standing alone explains even less to the reader why the authors came to this conclusion.

The methods section remains unclear: exact inclusion-/ exclusion criteria and the description of the intervention (e.g. which parameters were measured and which data collected at which points of time, what nutrition protocols and cut-offs were used…) are needed. In addition only in the discussion section it was mentioned that the funds were cut and changes to the initial plans had been made. In total, the lack of clarity and transparency leaves me quite unsatisfied.

In Abstract and Results section, exact descriptions are needed as well: numbers with p-values at are lacking most of the time (please see detailed comments below). Figures are missing, p-statistics for group comparisons are missing. The entire result section remains unclear.

As the manuscript is written now, in my opinion the methods and results do not support the authors conclusion, which seems to be “an elevated Total Energy Expenditure… would allow the human body to save more energy to fight off cancer”. This statement is contradicted in the manuscript itself, where it is stated that “Cancer patients were not hypermetabolic”. In addition an elevated TEE may lead to greater need for macronutrient intake.

The discussion lacks scientific citations to support the authors hypotheses and also a discussion and comparison of the obtained results within other trials of nutritional therapy in this patient group.

Specific comments:

Abstract:

Why was the median overall survival undefined? (also in Results section)

Methods: suggest to add some biomarkers and to state specifically which OS was used as primary endpoint

Results: Suggest to add specific numbers to BIA and TEE that were measured.

Introduction

“Nutritional status appears to be a major prog-nostic factor in oncologic patients, causing cancer-associated cachexia” – please rephrase this sentence – the nutritional status does not cause cachexia

“Metabolism of cancer patients …  and can be as high as 30 kcal/kg/day” – please revise the language

“The (ESPEN)….. , both diagnosis of cachexia and management remains low (10).” – this is also unclear -management of cancer remains low?

Suggest to specify what is meant by “tight caloric control” and give one sentence of background what was done before to enhance the readers understanding.

Materials & Methods

Study design& patients: Suggest to structure a bit more – first regulatory things/ registration, then details regarding patients etc.

Please provide one specific and concise section regarding in- and exclusion criteria

Also, please move results/ patient characteristics to the results section

Please move procedure-related descriptions to “procedures”

Was calorimetry performed in both groups?

“Survivors were followed for up to 12 months after the last surviving patient had entered the study.” This sentence is unclear, please rephrase.

What data were obtained by the mentioned sense-wear device, how does this device calculate energy expenditure and is this device validated to produce sensible results, especially when considering this cohort of quite ill patients.

Were possible changes in resting energy expenditure considered and measured within the intervention period?

Taking not even an approximate record of the macronutrient intake during and intensive intervention period over three months seems almost careless to me – especially since the authors state that oral intake was low at baseline. Please elaborate on why no supervision of the prescribed nutritional therapy was performed. In addition, a difference in nutritional intake between groups would add value to the performed intervention.

Please describe more clearly: What were the nutritional goals, was there an SOP regarding the steps of nutritional therapy? What were parameters influencing the choice of nutritional therapy and its changes

Results:

The first phrase belongs to the inclusion criteria.

The sentence about Karnofsky is redundand.

Please add numbers to your findings (medians, means, standard differences, p-values etc.)

Discussion

Please add scientific references to support your hypotheses and to compare your results to other studies of nutritional therapy in cancer patients.

There is a whole section of discussion about fat, which is not really related to the reported results.

“Increase in fat mass is not usually associated with survival in cancer patients, and both groups received the same daily recommendation of fat intake (see Ma-terials and Methods).” – there is a  lack of connection between the statements

Reviewer 2 Report

This study demonstrated that nutritional therapy based on biophysical measurements, rather than biochemical calculations, can improve overall survival in male cancer patients. The focus of this study is unique, however, several serious concerns make it difficult to accept.

Major points:

The backgrounds of the patients are so different that it is difficult to conclude that the nutrition intervention contributed to improved survival. The authors should pay more attention to the homogenization of the target patient backgrounds. Furthermore, this study did not correct for confounding by regression analysis. This is a very important omission.

There is a lack of documentation regarding energy sufficiency rates during the study period in both the nutrition intervention and control groups.

Minor points:

1. There are many errors in the Figure legends, making them difficult to understand.

2. There are many typographical errors in the text that need to be corrected.

3. Table 1 is very difficult to read and needs significant modification.

Reviewer 3 Report

General Comments

Cachexia is a critical clinical issue for patients with cancer. This paper makes an important contribution in building on the authors’ previous research regarding the potential of nutrition therapy to improve overall survival in patients with cancer cachexia and suggests a possible pathway.

Specific Comments

Title:  suggest inserting the word “proposed” before “chemical-physical pathway” since as identified in the abstract “the underlying mechanism appears to be related to“ and this single study is not definitive proof.

The Discussion concisely reviews the study results, considers a potential pathway, and identifies strengths and weaknesses of the study. The Discussion would benefit from additional content explaining how the study results and proposed pathway relate to other research in the literature as the Discussion currently does not include any research citations.